# Comparative Study on MNVT of OPV Type I and III Reference Products in Different Periods

**DOI:** 10.3390/diseases11010028

**Published:** 2023-02-08

**Authors:** Xiyan Wang, Ruirui Ren, Bo Ma, Jing Xie, Yan Ma, Hong Luo, Yu Guo, Ling Ding, Liang Zhang, Mengyuan Zhang, Tianlang Wang, Zhichao Shuang, Xiujuan Zhu

**Affiliations:** Beijing Institute of Biological Products Co., Ltd., Beijing 100176, China

**Keywords:** OPV, MNVT, neurovirulence, reference product, vaccine

## Abstract

Widespread vaccination using the oral live attenuated polio vaccine (OPV) and Sabin strain inactivated vaccine (sIPV) have greatly reduced the incidence of polio worldwide. In the period post-polio, the virulence of reversion of the Sabin strain makes the use of OPV gradually becoming one of the major safety hazards. The verification and release of OPV has become the top priority. The monkey neurovirulence test (MNVT) is the gold standard for detecting whether OPV meets the criteria, which are recommended by the WHO and Chinese Pharmacopoeia. Therefore, we statistically analyzed the MNVT results of type I and III OPV at different stages: 1996–2002 and 2016–2022. The results show that the upper and lower limits and C value of the qualification standard of type I reference products in 2016–2022 have decreased compared with the corresponding scores in the 1996–2002 period. The upper and lower limit and C value of the qualified standard of type III reference products were basically the same as the corresponding scores in the 1996–2002. We also found significant differences in the pathogenicity of the type I and III in the cervical spine and brain, with the decreasing trend in the diffusion index of the type I and type III in the cervical spine and brain. Finally, two evaluation criteria were used to judge the OPV test vaccines from 2016 to 2022. The vaccines all met the test requirements under the evaluation criteria of the above two stages. Based on the characteristics of OPV, data monitoring was one of the most intuitive methods to judge changes in virulence.

## 1. Introduction

Poliomyelitis is an acute infectious disease caused by poliovirus (PV), which seriously endangers children’s health [1,2]. The virus causes motor neuron cell damage by invading the central nervous system, mostly affecting motor neurons in the anterior horn of the spinal cord, leading to acute flaccid paralysis (AFP) with muscle atrophy. Its wide spread in the world has led to paralysis, or even death, of countless children, or life-long affliction with post-polio syndrome (PPS) [3,4,5,6].

There are three serotypes of poliovirus: poliovirus type I, poliovirus type II and poliovirus type III. In 2015, the WHO announced that wild-type poliovirus type II was completely eradicated and type I and III strains became a priority for prevention [7]. Therefore, mass vaccination with vaccines became an important means of preventing poliomyelitis. The data evaluation model shows that after widespread vaccination, 5 million additional cases of paralytic poliomyelitis were prevented during 1960–1987, and 24 million cases were prevented globally during 1988–2021 [8].

Inactivated poliovirus vaccine (IPV) and live attenuated oral poliovirus vaccine (OPV) are the important methods to prevent poliomyelitis from happening. The intestinal immune effect induced by OPV can better prevent the transmission of wild virus compared with IPV, but at the same time, it is accompanied by a virulence of regression phenomenon higher than IPV [9]. To better prevent the spread of the poliovirus, comprehensive vaccination has become a necessary method. However, large-scale vaccination means that manufacturers need to strictly enforce quality control for vaccines to reduce the incidence of clinically adverse reactions.

In the process of vaccine production, we need to complete a large number of experiments to provide data support for the verification of vaccine safety and effectiveness. MNVT is used to detect the virulence of seed virus strains and OPV stock solution, and then determine whether the vaccine virulence is qualified. This test is crucial in the vaccine inspection process. In this paper, MNVT experiment results of OPV production process at the Beijing Institute of Biological Products Co., ltd. (Beijing Companies, Beijing, China) were statistically compared to provide a basis for vaccine quality control.

MNVT has high repeatability and sensitivity in virulence testing with a half-century history of application, which is an essential step in evaluating the safety of live attenuated polio vaccine [10]. In accordance with the WHO and the Chinese Pharmacopoeia, the animals were infected with virus by intraspinal injection or intracerebral injection, the glia in the central nervous system (CNS) was activated, and the peripheral immune cells infiltrated and spread to the whole CNS. The semi quantitative score was made by observing the pathological changes to evaluate the batch release of live attenuated vaccines [11].

In the 1980s, fixed upper and lower limits and C values were set as criteria for determining whether the MNVT was acceptable. In the past, we have completed the statistics of MNVT data for three types of reference products from 1996 to 2002, and verified the differences between the experimental batch’s MNVT results and those from before the 1980s [12]. Each laboratory should establish its own judgment standard, recommended by the WHO, after completing experiments with the continuous accumulation of experimental data. Therefore, after 2013, the M value of the first ten experiments and the C value calculated were used by combining the standard deviations as the evaluation criteria for vaccine eligibility.

In this paper, the MNVT results of type I and type III were counted using references from 2016–2022 and compared with the results of 1996–2002. With respect to pathological test results, the mean fraction of lesions at neutrophilic sites and C values were determined. Furthermore, there were certain differences in the criteria for determining MNVT in different time periods, which was used to further improve the OPV MNVT test results and provide a certain database for subsequent experiments.

## 2. Materials and Methods

### 2.1. Animals

According to the requirements of the Chinese Pharmacopoeia and WHO regulations, we selected Macaca mulatta (over 1.5 kg) as the test animal, which were provided by the Beijing Institute of Xieerxin Biology Resource (Beijing, China) and Xiangcheng Longrui Experimental Animal Co., Ltd. (Henan, China). We selected monkeys before the test and isolated them for 6 weeks. After blood sampling and testing, we ensured that the monkeys did not carry tuberculosis, B virus, foam virus (FV) or other acute infectious diseases, and there were no neutralizing antibodies against poliovirus in serum. Type I and type III were immunized in 14 and 22 heads of monkeys, respectively, to ensure that the effective number of monkeys after the test was not less than 11 and 18.

All animals were housed in a room maintained at 16–26 °C with an alternating 12 h light/dark cycle (AM. 7:00–PM. 19:00). Food and water were autoclaved. The operation process of animal experiment conforms to the national Regulations on the Administration of Experimental Animals and has been reviewed by the Ethics Committee of Experimental Animal Welfare of Beijing Company. The manufacturers of experimental animals have obtained the production license of experimental animals approved by the Beijing Municipal Science Committee and the domestication and breeding license of wild animals approved by the Beijing Landscaping Bureau.

All efforts were made to minimize animal pain and suffering and the number of animals used during the experiments.

### 2.2. Materials

The reference product of type I and type III provided by the WHO (I—1981, III—1981), the monovalent virus stock solution of OPV was provided by Beijing companies.

### 2.3. Methods

MNVT is widely used in the detection of attenuated live vaccines. As an important standard for judging neurovirulence, this method has been used in vaccine quality control in the pharmaceutical industry for more than 50 years. Since 1996, all the experimental steps were carried out in accordance with the requirements of the WHO Regulations for the Manufacture and Testing of Live Attenuated Oral Polio Vaccine and the Pharmacopoeia of the People’s Republic of China (current edition).

The experiment was divided into a vaccine group to be tested and a reference vaccine group. The number of effective monkeys in each group of type I should be more than 11, and the number of effective monkeys of type III should be more than 18. The group of reference and vaccine to be tested should be carried out in parallel. All animals were injected between the first and second lumbar vertebrae, and each monkey was injected with 0.1 mL of sample (the virus content should be 6.5~7.5 LgCCID50/mL).

Spinal cord injection can make the virus directly invade the central nervous system (CNS) tissue. After vaccination, it is observed for 17–22 consecutive days, and the daily feed intake (normal; feed intake 1/2; feed intake 1/3; feed intake waste), fecal conditions (fecal formation; fecal beach; fecal porridge) and motor status (normal climbing; hind limbs unable to move; limbs unable to move) were recorded. If an animal dies during clinical observation, the animal shall be dissected to further confirm the cause of death, and the number of animals killed during observation shall not exceed one quarter before the experiment can be established.

At the end of observation period, sections of the CNS of monkeys were taken for histological examination, and the thickness of the sections was 10–15 μm. Five pathological sections were prepared from each animal, resulting in a total of 29 tissues: 10 tissues from the cervical enlargement, 12 tissues from the lumbar enlargement, 1 from the cerebellum, 1 from the pons, 2 from the medulla oblongata, 2 from the midbrain, 2 from the cortex, and 2 from the thalamus. The specific distribution is as follows:

Slice 1: Swollen neck, 10 slices.

Slice 2: Waist puffed, 12 slices.

Slice 3: 1 section of cerebellum and pontine and 2 sections of medulla oblongata.

Slice 4: Midbrain and cortex, 3 slices.

Slice 5: Thalamus, 1 slice.

After fixation, staining and depigmentation, the pathological sections were prepared, and the lesions in each monkey were observed by microscopic examination and counted using a 4-level scoring method. The scoring process was performed independently by the same experimenter and criteria were as follows:

Score 1: Only cellular invasion like perivascular cufflike leukocyte aggregates; Low, moderate or high cellular invasion by non-neural lesions (which alone would not be sufficient to indicate a positive monkey).

Score 2: Cellular infiltration and little neuronal damage.

Score 3: Cell invasion and extensive neuronal damage.

Score 4: Massive neuronal damage with or without cell invasion.

Of concern is that monkeys with neuronal damage in slices, but without a needle track should be considered valid monkeys. Trauma-induced damage in sections without specific pathological changes was not considered valid. Sections could not be included in the scoring if the damage on the sections was a result of trauma and not specific viral damage. Pathological sections were scored by scoring method to determine the activity of virus-induced neuropathy. When the mean value of the reference vaccine was between the upper and lower limits, the vaccine to be tested can be judged to be qualified based on the respective standard values. If the mean value of the vaccine to be tested exceeds the upper and lower limits, the experimental result does not hold.

The mean value of the reference lesion, the total within test error (s2) and combined sample standard deviation (s) were calculated using the sampling error of the statistical mean. The upper limit of type I was calculated as M+s, the lower limit was calculated as M-s, the upper limit of type III was calculated in the same way as type I, and the lower limit comes from M-s/2.

The acceptability constant (C value) was the difference between the average lesion score of the vaccine to be tested and the average lesion score of the reference vaccine. The calculation method is as follows: $C_{1}=2.3\sqrt{2S^{2}/N_{1}}$, $C_{2}=2.6\sqrt{2S^{2}/N_{1}}$, $C_{3}=1.6\sqrt{2S^{2}/N_{2}}$. The results were judged as follows: the average lesion score (${\overset{-}{x}}_{\mathrm{test}}$) of the vaccine to be tested was compared with the average lesion score (${\overset{-}{x}}_{\mathrm{ref}}$) of the reference product, if ${\overset{-}{x}}_{\mathrm{test}}-{\overset{-}{x}}_{\mathrm{ref}}<C_{1}$, the vaccine was qualified, otherwise it is unqualified; $C_{1}{<\overset{-}{x}}_{\mathrm{test}}-{\overset{-}{x}}_{\mathrm{ref}}<C_{2}$ requires it to be retested and recalculated after the retest. If ${\overset{-}{x}}_{\left(\mathrm{test}1+\mathrm{test}2\right)}-{\overset{-}{x}}_{\left(\mathrm{ref}1+\mathrm{ref}2\right)}/2>C_{3}$, it will be judged unqualified.

## 3. Results

### 3.1. Statistics of Pathological Test Results of Type I MNVT

The results of 19 tests were counted using references from 2016 to 2022 and it was found that the pathological score of the reference products fluctuated greatly, the upper and lower limits gradually narrowed (Figure 1); there were three consecutive batches of results approaching the lower limit, which further narrows the upper and lower limits. Compared with 1996–2002, it could be more intuitively found that the mean value of the 19 tests (0.383) decreased by 0.222, the upper limit (0.591) decreased by 0.392 and the lower limit (0.231) decreased by 0.166; the value has a relative increase of 0.018, with no significant change (Table 1).

### 3.2. Statistics of Pathological Test Results of Type III MNVT

A total of 12 trials of type III reference products were completed from 2016–2022. The results demonstrated that the pathological score of the reference product fluctuated from 2016 to 2022, the upper and lower limits gradually narrowed down (Figure 2). Different from the changes of type I, the pathological data of type III references have decreased to varying degrees compared with the past. The average mean value of the 12 experiments (0.538) was decreased by 0.194, the upper limit (0.940) was 0.211 lower than the past average (1.151) and just the lower limit (0.440) was improved by 0.131 compared to the past (0.309) (Table 2). Although all were within the qualified range, the results still require our close attention.

### 3.3. Comparison of Lesion Scores in Different Neurotropic Sites of Type I and Type III

The pathological scores of three parts of all reference products from 2016 to 2022 were counted. As the results displayed (Figure 3), the degree of lesions in the enlarged lumbar region was still significantly higher than that in the cervical spine and brain. There was no significant difference between type III and type I lumbar lesions (*p* > 0.05), although the cervical spine and brain lesions index was significantly higher than type I (*p* < 0.001). As before, we compared the statistical data from 1996 to 2002 (Table 3), and the overall lesion score decreased. The coefficient of variation (CV) was calculated based on the standard deviation value of the previous ten times. Compared with the 1996–2002 period, we found that the coefficient of variation of type I has increased, and that of type III decreased, indicating that the degree of dispersion of pathological scores of the two reference products was inconsistent.

The average pathological scores of the lumbar spine, neck and brain marrow of the reference article are the same as the previous ones.

The statistical data from 1996 to 2002 was compared (Table 3), and the overall pathological score decreased. The coefficient of variation (CV) value was calculated based on the standard deviation of ten/the average M value of ten. Compared with 1996–2002, we found that the coefficient of variation of type I increased, and type III decreased, so it was reflected that the dispersion degree of the two references were inconsistent. At the same time, the extramedullary diffusion index in the two periods were compared. The lumbar spinal cord was the part inoculated in the MNVT, and the motor neurons in the anterior horn were also the most concentrated, the lesions of the lumbar spine were the most serious compared with the cervical spine and the brain. Although the diffusion index of the two to the neck and brain was not much different between the two periods, there was still a slight downward trend now compared with the past. It was also consistent with the change trend of the pathological mean.

### 3.4. C Value Statistic

The C value (C1, C2, C3) represents the acceptable range of the difference between the vaccine test and the reference lesion score, and the change of the C value directly affects whether the vaccine was qualified or not. We counted the changes of C value in the above two stages. The data illustrated that (Table 4) the C value of type I and type III decreased. The results displayed that although the virulence of vaccines has declined in recent years, the qualification limit of the test vaccines has gradually become stricter, the pass rate of the vaccines has decreased and the difficulty has increased.

### 3.5. MNVT Results of Test Vaccines in 2016–2022

The MNVT results of type I (Table 5) and III (Table 6) OPV stock solution tested were collected by all Beijing companies from 2016 to 2022 and calculated based on the results of the same batch of reference products. The virulence of all the test vaccines were all within the qualified range. In addition, we also compared with the fixed C1 value at the past stage and found that the results of the test vaccines were still qualified under different reference values, which further verifies the reliability of the calculation method at this stage.

## 4. Discussions

Although the Global Polio Eradication Initiative (GPEI) promised to eradicate polio by 2000, the disease has remained endemic in some countries for the past 20 years [13]. However, the incidence of poliomyelitis in the world has been greatly reduced, the crucial reason for this is vaccination.

OPV are one of the most successful methods for controlling PV infections, the intestinal mucosal immunity and systemic immunity caused by OPV make the immune effect much higher than that of IPV, which could more effectively prevent the spread of wild poliovirus (WPV). However, because of outbreaks associated with circulating vaccine-derived poliovirus (cVDPVs) [14,15,16], most people have recognized that the risk of OPV reversion of virulence is significantly higher than that of IPV. Based on the difference in immune efficacy between the OPV and IPV, complete cessation of OPV use is still not possible. Instead, because of the continued emergence of cVDPVs, OPV production and use potentially needs to increase progressively. In 2016, the WHO recommended at least one dose of IPV preceding routine immunization with OPV vaccination to reduce vaccine-associated paralytic polio (VAPPs) and VDPVs until PV could be eradicated [17,18,19]. We have completed the transformation from tOPV to bOPV before and have been using it until now, making great contributions to the world’s prevention of poliomyelitis, which also means that we have greater responsibility. The significance of our comparison and analysis of the MNVT results for the reference product in the above two stages is to further ensure whether the virulence of OPV is continuously applicable.

Although MNVT is the gold standard for detecting vaccine neurotoxicity, it cannot explain many potential neurotoxicity mechanisms and lacks reproducibility [20,21]. The WHO also recommended detection methods other than MNVT. We found that the detection of spontaneous neurotoxic response of OPV by PCR and restriction endonuclease cleavage (MAPREC) is highly sensitive and reliable. This method can predict the experimental results of MNVT [22,23]. We also completed the quantitative analysis of viruses in OPV through MAPREC, further ensuring the full tracking of virulence [24], which is used for the safety and consistency control of OPV together with MNVT. Although the application of MNVT and MAPREC is mature enough, new alternative methods are still crucial for considering the 3Rs (Replacement, Reduction and Refinement) and methods optimization.

A correlativity study showed that transgenic mice maybe are one of the best alternative models. PVR-Tg21 transgenic animals developed by Japanese scholars in 1990 indicated the same virus sensitivity as MNVT [25,26,27]. We are currently doing more work to try to determine the feasibility and sensitivity of transgenic mouse models. In addition to in vivo methods, new in vitro molecular diagnosis is still the focus of attention in the future, we need most effort to further ensure the absolute safety and effectiveness of each vaccine.

## 5. Conclusions

To further ensure whether the virulence of OPV vaccine is continuously applicable, the results of MNVT experiments over the past decades were counted, compared and analyzed. The statistical results illustrated that the average lesion score of type I was on a downward trend, which was different from that in the 1996–2002 period. Compared with the same trend, the C value has also decreased, which suggests that the qualification rate of our vaccine may be affected. However, compared with the changes of type I, type III maintained relatively stable virulence in the above two stages. The reason of decreasingly pathological value of type I also needs to be further investigated.

In addition to the reference products, we have counted the MNVT results of OPV vaccine stock solution from 2016 to 2022. The data showed that the results met the test requirements under different evaluation criteria in 1996–2002 and 2016–2022. Nevertheless, we also need to constantly monitor the virulence changes of reference products to strictly control the qualification rate of vaccines.

## Figures and Tables

**Figure 1 diseases-11-00028-f001:**
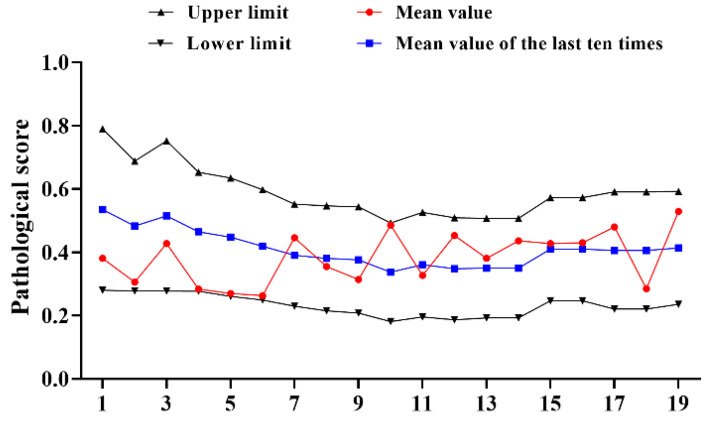
Statistical of pathological score of type I reference product in 2016–2022. N = 19 times.

**Figure 2 diseases-11-00028-f002:**
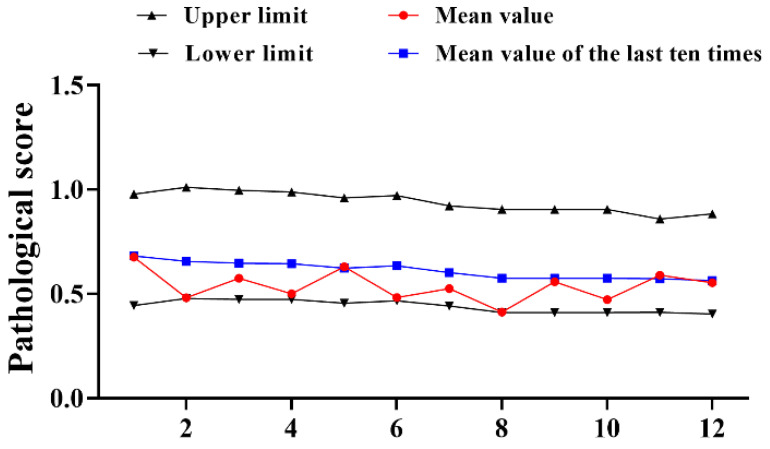
Statistical of pathological score of type III reference product in 2016–2022. N = 12 times.

**Figure 3 diseases-11-00028-f003:**
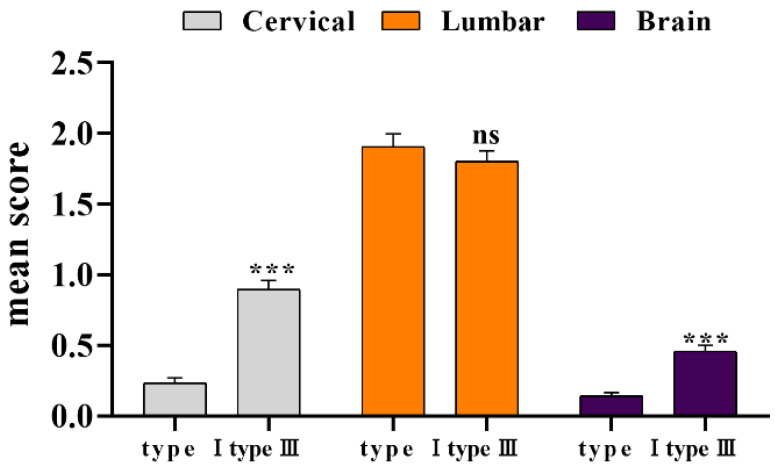
Statistical of pathological lesion tissue of type I and type III. *** *p* < 0.01 compared with type I group.

**Table 1 diseases-11-00028-t001:** Statistics for two stage of type I.

Stage	1996–2002	2016–2022	Increase	Decrease
Mean value	0.605	0.383	-	0.222
Combined sample standard deviation (s)	0.161	0.179	0.018	-
upper limit	0.591	0.938	0.347	-
lower limit	0.397	0.231	-	0.166

**Table 2 diseases-11-00028-t002:** Statistics for two stage of type III.

Stage	1996–2002	2016–2022	Increase	Decrease
Mean value	0.732	0.538	-	0.194
Combined sample standard deviation (s)	0.423	0.331	-	0.092
upper limit	1.151	0.940	-	0.211
lower limit	0.309	0.440	0.131	-

**Table 3 diseases-11-00028-t003:** Statistics of the average lesion scores in different central nervous systems from 1996–2002 and 2016–2022.

CNS	Type I		Type III	
	1996–2002	2016–2022	1996–2002	2016–2022
Cervical	0.276	0.118	0.532	0.449
Lumbar	1.404	0.953	1.187	0.900
Brain	0.127	0.070	0.473	0.229
Coefficient of variation (cv)	0.346	0.437	0.851	0.543
Extramedullary diffusion index	0.223	0.167	0.459	0.429

**Table 4 diseases-11-00028-t004:** 1996–2002 and 2016–2022 C value statistics.

C Value	1996–2002	2016–2021	Increase	Decrease
Type I C1	0.224	0.160	-	−0.064
Type I C2	0.253	0.180	-	−0.073
Type I C3	0.110	0.081	-	−0.029
Type III C1	0.282	0.243	-	−0.039
Type III C2	0.318	0.274	-	−0.044
Type III C3	0.139	0.123	-	−0.016

**Table 5 diseases-11-00028-t005:** The mean value of type I OPV stock solution between 2016 and 2022 and the MNVT qualification under different judgment standards.

NO.	Positive Monkeys	Mean_test_	Mean_ref_	Mean_test_–Mean_ref_	2016–2022	1996–2002 (0.224)
1	13	0.461	0.381	0.08	qualified	qualified
2	13	0.279	0.306	−0.027	qualified	qualified
3	14	0.449	0.428	0.021	qualified	qualified
4	14	0.407	0.284	0.123	qualified	qualified
5	14	0.273	0.270	0.003	qualified	qualified
6	14	0.266	0.263	0.003	qualified	qualified
7	14	0.360	0.446	−0.086	qualified	qualified
8	14	0.476	0.355	0.121	qualified	qualified
9	14	0.268	0.314	−0.046	qualified	qualified
10	14	0.449	0.486	−0.037	qualified	qualified
11	13	0.286	0.327	−0.041	qualified	qualified
12	13	0.417	0.453	−0.036	qualified	qualified
13	14	0.381	0.381	0	qualified	qualified
14	14	0.491	0.436	0.055	qualified	qualified
15	14	0.444	0.428	0.016	qualified	qualified
16	12	0.398	0.43	−0.032	qualified	qualified
17	13	0.552	0.48	0.072	qualified	qualified
18	13	0.399	0.285	0.114	qualified	qualified
19	14	0.429	0.529	−0.1	qualified	qualified

**Table 6 diseases-11-00028-t006:** The mean value of type III OPV stock solution between 2016 and 2022 and the MNVT qualification under different judgment standards.

NO.	Positive Monkeys	Mean_test_	Mean_ref_	Mean_test_–Mean_ref_	2016–2022	1996–2002 (0.282)
1	22	0.712	0.676	0.036	qualified	qualified
2	21	0.531	0.481	0.05	qualified	qualified
3	21	0.544	0.575	−0.031	qualified	qualified
4	20	0.382	0.500	−0.118	qualified	qualified
5	20	0.468	0.630	−0.162	qualified	qualified
6	22	0.536	0.482	0.054	qualified	qualified
7	21	0.542	0.525	0.017	qualified	qualified
8	22	0.582	0.413	0.169	qualified	qualified
9	21	0.478	0.558	−0.08	qualified	qualified
10	22	0.647	0.472	0.175	qualified	qualified
11	21	0.619	0.590	0.029	qualified	qualified
12	21	0.524	0.553	−0.029	qualified	qualified

## Data Availability

Not applicable.

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
