# Peer review of "Comparative Study on MNVT of OPV Type I and III Reference Products in Different Periods"

_diseases, 2023, doi:10.3390/diseases11010028_

Round 1

Reviewer 1 Report

In the present manuscript, Wang Xiyan and collaborators present interesting and very useful results nowadays because of the emergence of cases of poliomyelitis derived from vaccine strains. 

 Is a very well-designed and written manuscript. I only suggested some details in order to enrich the ideas and the results. 

I wonder if the authors have any statistical data regarding the number of cases of AFP that could be associated with the vaccines evaluated in this paper, maybe any number of cases reported in the periods referenced, independently that are very much factors (epigenetic) to study. 

Please reference in the text the last two tables (5 and 6) and I think that both titles could be improved. 

Author Response

I wonder if the authors have any statistical data regarding the number of cases of AFP that could be associated with the vaccines evaluated in this paper, maybe any number of cases reported in the periods referenced, independently that are very much factors (epigenetic) to study. 

Re:We have verified that the relevant batches of vaccines mentioned in the article have not caused serious adverse reactions or AFP in the past, so we have not described them in this paper.

Please reference in the text the last two tables (5 and 6) and I think that both titles could be improved.

We re-evaluated and revised the table 5 and 6 in the manuscript.

Table 5 :  The Mean Value of type I OPV stock solution between 2016 and 2022 and the MNVT qualification under different judgment standards

Table 6:  The Mean Value of type III OPV stock solution between 2016 and 2022 and the MNVT qualification under different judgment standards

Thank you very much for your valuable comments.

Reviewer 2 Report

Xiyan, W. et al. wrote, "Comparative study on MNVT of OPV type I and III reference products in different periods.”  The manuscript emphasized the periodical evaluation of the vaccines, which is essential to evaluate the quality of the vaccines.

Minor corrections:

1.     Live attenuated oral polio vaccine, which is not a live attended oral polio vaccine, may be typing error in the third paragraph of the introduction.

2.     2.1 animal paragraph needs to be re-written; it looks like it is written for the project proposal.

3.     Reviewer wondered if there was any possibility of virus mutation between 1996-2002 vs. 2016-2022, which could be the possible reason for narrowing the pathological data compared with the past, as in figure 2.

4.     Figure 3 x-axis labeling can be retyped to reflect the particular bar.

Author Response

 Live attenuated oral polio vaccine, which is not a live attended oral polio vaccine, may be typing error in the third paragraph of the introduction.

Re:The writing error has been corrected in the manuscript. Please check the revised manuscript for details.

  1. 2.1 animal paragraph needs to be re-written; it looks like it is written for the project proposal.

Re: Thank you very much for your valuable views. We have revised the description of the animal paragraph. Please refer to the revised manuscript for details.

  1. Reviewer wondered if there was any possibility of virus mutation between 1996-2002 vs. 2016-2022, which could be the possible reason for narrowing the pathological data compared with the past, as in figure 2.

Re:We have not found any significant changes in the toxicity of the reference products, until now. However, since the type I and type III reference products used in the test from 1996 to 2022 are the same batch provided by WHO. The only difference is that the storage time of reference products in the two stages is different. We will continue to pay attention to the changes in the change of this batch to ensure that the OPV virulence test will not be affected.

  1. Figure 3 x-axis labeling can be retyped to reflect the particular bar.

Re:I'm sorry that this is an omission in the writing process. We have modified and replaced the error picture.

Reviewer 3 Report

Major Suggestions:

In this manuscript, authors presented well defined and scientifically sound research regarding the MNVT studies of type I and III OPV at different periods (1996-2002 and 2016-2022).

The Title of the manuscript is concise and relevant. The aim and scope of the study explained well. The abstract is easy to understand and well written. Introduction is quite comprehensive and highlighted work importance. Materials and methods are quite descriptive. Authors provided well explained interpretation of results and discussion. Overall, this research article is nicely written.

Before proceeding further, I expect the authors to thoroughly proofread the document and fix all grammatical and typographical errors (some examples include L33, L67, L89, L213, L226, L233, L238 etc).

Also, the back matter (FundingAcknowledgements, Author Contributions, Conflicts of Interest, Institutional Review Board Statement) in the manuscript was found missing. Kindly go through the author’s instructions and provide necessary information.

Author Response

  1. Before proceeding further, I expect the authors to thoroughly proofread the document and fix all grammatical and typographical errors (some examples include L33, L67, L89, L213, L226, L233, L238 etc).

Re: We re-read and correct the grammatical and typographical errors, and sincerely thank you for your comments.

  1. Also, the back matter (Funding, Acknowledgements, Author Contributions, Conflicts of Interest, Institutional Review Board Statement) in the manuscript was found missing. Kindly go through the author’s instructions and provide necessary information.

 Re: Sorry about that we missed this part of information. We have completed the supplement in the manuscript. Thank you for your correction.

Round 2

Reviewer 3 Report

The manuscript has been revised as per the suggestions, and done with all the necessary correction.

Minor Suggestion: Please provide date of approval in ‘Institutional Review Board Statement’ (refer author’s instructions).  

Author Response

Minor Suggestion: Please provide date of approval in ‘Institutional Review Board Statement’ (refer author’s instructions).  

Re: Institutional Review Board Statement: All experiments have passed the review of the Ethics Committee of Laboratory Animal Welfare of Beijing Institute of Biological Products Co., Ltd (The first approval date was July 1, 2014, before which there were no welfare ethics approval requirements for animal experiments in China.)

Thank you very much for your comments, and we have supplemented the earliest approval pass time at the end of the manuscript. It is important to note here that China requested that the time for ethical review of welfare was 2014, but we did not have this approval before 2014 due to our longer statistical time for our data.